# Discovery of hexagonal ternary phase $Ti_2InB_2$ and its evolution to layered boride TiB

Junjie Wang [1,2,3,4], Tian-Nan Ye[3,4], Yutong Gong [3], Jiazhen Wu [3], Nanxi Miao[1,2], Tomofumi Tada [3] & Hideo Hosono[3]

$M_{n+1}AX_n$ phases are a large family of compounds that have been limited, so far, to carbides and nitrides. Here we report the prediction of a compound, $Ti_2InB_2$, a stable boron-based ternary phase in the Ti-In-B system, using a computational structure search strategy. This predicted $Ti_2InB_2$ compound is successfully synthesized using a solid-state reaction route and its space group is confirmed as $P\bar{6}m2$ (No. 187), which is in fact a hexagonal subgroup of $P6_3/mmc$ (No. 194), the symmetry group of conventional $M_{n+1}AX_n$ phases. Moreover, a strategy for the synthesis of MXenes from $M_{n+1}AX_n$ phases is applied, and a layered boride, TiB, is obtained by the removal of the indium layer through dealloying of the parent $Ti_2InB_2$ at high temperature under a high vacuum. We theoretically demonstrate that the TiB single layer exhibits superior potential as an anode material for Li/Na ion batteries than conventional carbide MXenes such as $Ti_3C_2$.

[1] State Key Laboratory of Solidification Processing, Northwestern Polytechnical University, Xi'an 710072 Shaanxi, People's Republic of China. [2] International Center for Materials Discovery, School of Materials Science and Engineering, Northwestern Polytechnical University, Xi'an 710072 Shaanxi, People's Republic of China. [3] Materials Research Center for Element Strategy, Tokyo Institute of Technology, 4259 Nagatsuta-cho, Midori-ku, Yokohama, Kanagawa 226-8503, Japan. [4] These authors contributed equally: Junjie Wang, Tian-Nan Ye. Correspondence and requests for materials should be addressed to J.Wu. (email: wujzphystu@gmail.com) or to H.H. (email: hosono@mces.titech.ac.jp)

**M**n+1AXn (MAX) phases are a class of unique materials that exhibit a combination of ceramic and metallic properties, and a mixture of covalent and metallic bonding[1–3]. Therefore, MAX phases possess the features of elastically stiff, strong, and heat-tolerant ceramics[4], although their electrical and heat conductivities drop linearly with increasing temperature, as with a metal[5]. For the reported MAX phases, M represents an early transition metal, A is generally a metal element in group 13 or 14, while X is limited to C or N. Utilizing the significant difference in strength between the metallic M–A bonding and covalent M–X bonding, the A-layer can be selectively etched to form two-dimensional (2D) materials known as MXenes, which cannot be synthesized directly due to their thermodynamic metastability[6–9]. MAX phases and the derived MXenes have attracted extensive interest because of the abundance of their possible forms and structures, and their excellent chemical and mechanical stabilities, and thus, their broad applicability[10–16]. For instance, MXenes have shown great promise in applications involving electrochemical energy storage[10–14] and catalysis[15,16] because of their excellent conductivity, rich interlayer porosity and high surface functionalization. However, the question has been raised as to whether it is possible to synthesize new MAX phases and corresponding MXene materials without the limitation of using C and N as the X component.

Recently, with the successful deposition of atomic boron layers on various metal surfaces[17,18], 2D boron and related borides have attracted broad attention for their potential applications in nanoelectronic devices[19–21]. Therefore, the exploration of boron (B)-containing MAX phases and their derived MXenes with exotic properties is of considerable interest in this research field. Ade and Hillebrecht recently proposed several ternary borides as analogs of MAX phases by introducing B as the X element[22]. However, each boride presented in their work was orthorhombic, which is totally different from the hexagonal structures ($P6_3/mmc$ symmetry) of known MAX phases. In MAX phases, the M, A, and X atoms alternately stack along a hexagonal close-packed (HCP) manner and respectively form equilateral triangles of their own (the equilateral nature is determined by the symmetry of the hexagonal space group) parallel to each other. However, M and A atoms in the reported borides alternately stack along orthorhombic manner. And the M atoms, which are coordinated with boron, form non-equilateral trigonal prisms, with the side edge along $x$ direction determining the lattice constant $a$, perpendicular to the A layers. Furthermore, the nearest neighbor boron atoms in reported borides form one-dimensional zig-zag chains perpendicular to the A layers also. Therefore, these B-containing ternary compounds were not categorized as new MAX phases, but were instead named MAB phases. However, Ade and Hillebrecht did indicate the possibility of extending the family of MAX phases to B-containing compounds. However, it is difficult to explore ternary compounds using conventional techniques because the number of candidates is too large (about 100,000 compounds), even without considering the exponentially increasing number of reactions involved. Therefore, a feasible strategy to simplify the search for ternary compounds based on the available domain knowledge is in high demand.

Recent investigations[20,21] have revealed that an abundance of stable or metastable 2D structures can be formed using Ti and B. Our preliminary study and ref. [21] both suggested that TiB and Ti3B4 can exist in the form of layered structures in which TiB (space group $Cmcm$) or Ti3B4 (space group $Immm$) layers are connected by Ti-Ti metallic bonds (See Fig. 1 and Supplementary Fig. 1). The predicted Ti3B4 shows the same structure as that reported in the previous experiments[23]. However, the reported TiB compound in experiment[24] possesses the space group of

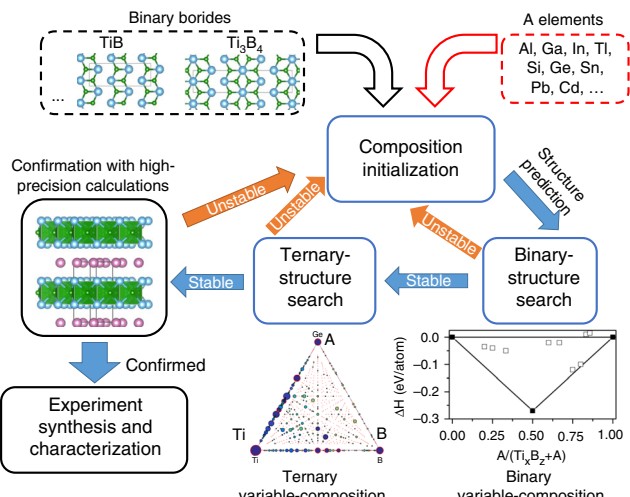

**Fig. 1** Calculation-based approach to the discovery of novel boron-containing ternary phases. The silver blue, purple, and green spheres in the structure models indicate Ti, In, and B atoms, respectively

$Pnma$, indicating that a direct formation of layered TiB ($Cmcm$) is prohibited. Therefore, a possible strategy could begin from available layered TiB and Ti3B4 and involve inserting A elements into the interlayer Ti-Ti bonds to synthesize B-containing MAX phases, with subsequent etching of A to obtain the related MXenes. Although TiB and Ti3B4 are metastable compounds, the synthesis of a TixAyBz could be possible if the reaction enthalpy is negative with respect to TixBz (TiB2, TiB and Ti3B4), bulk A and other competitive phases.

In the present work, we examine the possibility of forming boron-containing MAX phases by employing a computation-based strategy. The current strategy involves greatly improving the computation efficiency to identify Ti2InB2 and Ti2SnB2 as theoretically stable compounds at ambient pressure and a high pressure of 10 GPa, respectively. Based on theoretical predictions, Ti2InB2 was successfully synthesized through a solid-state reaction, and subsequently a layered TiB compound was obtained by the removal of indium through a high-temperature dealloying process. The obtained compounds were evaluated experimentally and were confirmed to be consistent with the predictions. The layered TiB was predicted to be a promising anode material for Li/Na ion batteries according to density functional theory (DFT) calculations. Pristine TiB monolayers possess a much higher ion (Li or Na) storage capacity than Ti3C2, a representative MXene, and exhibit a very low energy barrier for the diffusion of Li and Na ions.

## Results

**Stability and structures of predicted borides.** Figure 1 shows the thermodynamic stability of possible TixAyBz compounds estimated through reactions of TixBz with A rather than using Ti, B, and A. In the present material design loop, this step was realized by conducting binary variable-composition searches as implemented in the USPEX code[25–28] using TixBz and bulk A as ending compounds. A pre-investigation to identify the structure of TixBz was performed by employing a binary structure search in the Ti-B system (Supplementary Fig. 1). For the given types of TixBz and elemental A, each possible combination of TixBz and A was considered in the preliminary structure search, limited only by the total number of atoms per unit cell. The elements Al, Ga, In, Tl, Si, Ge, Sn, Pb, and Cd were considered as A candidates in the structure search. When one ternary compound, TixAyBz, was found to be thermodynamically stable with respect to the end

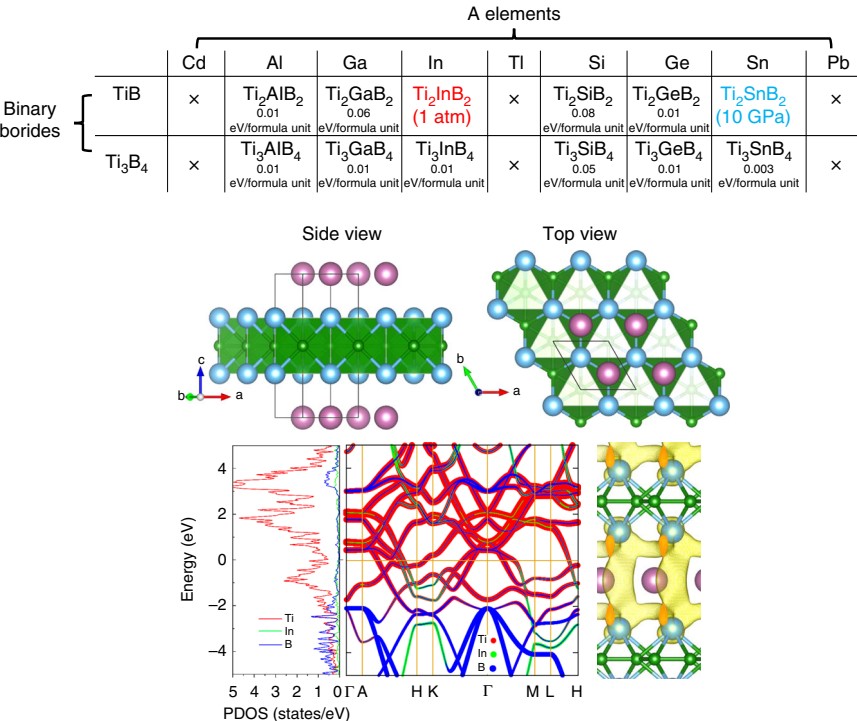

| | | Cd | Al | Ga | In | Tl | Si | Ge | Sn | Pb |
|---|---|---|---|---|---|---|---|---|---|---|
| | | | | | A elements | | | | | |
| Binary borides | TiB | × | $Ti_2AlB_2$ 0.01 eV/formula unit | $Ti_2GaB_2$ 0.06 eV/formula unit | $Ti_2InB_2$ (1 atm) | × | $Ti_2SiB_2$ 0.08 eV/formula unit | $Ti_2GeB_2$ 0.01 eV/formula unit | $Ti_2SnB_2$ (10 GPa) | × |
| | $Ti_3B_4$ | × | $Ti_3AlB_4$ 0.01 eV/formula unit | $Ti_3GaB_4$ 0.01 eV/formula unit | $Ti_3InB_4$ 0.01 eV/formula unit | × | $Ti_3SiB_4$ 0.05 eV/formula unit | $Ti_3GeB_4$ 0.01 eV/formula unit | $Ti_3SnB_4$ 0.003 eV/formula unit | × |

**Fig. 2** Results of theoretical structure search. **a** Summary of structure search results using binary and ternary variable-composition methods, where the distances for metastable phases to the convex hull are labelled in the unit of eV per formula. **b** Crystal structure. **c** Calculated electronic structure for stable boron-containing ternary phase $Ti_2InB_2$. The projected density of states (DOS), projected band structure and partial charge map on the (110) plane ($-0.5\,eV < E - E_f < 0.5\,eV$) for $Ti_2InB_2$ are shown in the left, center, and right panels, respectively

compositions of $Ti_xB_z$ and A in the binary variable-composition search, a ternary variable-composition search was started for this Ti-A-B system to estimate the global stability of the predicted $Ti_xA_yB_z$ structure. Otherwise, the search procedure was initialized again by the selection of different pairs of $Ti_xB_z$ and A elements.

Employing the strategy shown in Fig. 1 resulted in the prediction of a series of ternary compounds with the formulas $Ti_2AB_2$ and $Ti_3AB_4$ in the preliminary structure search (Fig. 2a). The ternary compounds $Ti_2AlB_2$, $Ti_3AlB_4$, $Ti_2GaB_2$, $Ti_3GaB_4$, $Ti_2InB_2$, $Ti_3InB_4$, $Ti_2SiB_2$, $Ti_3SiB_4$, $Ti_2GeB_2$, $Ti_3GeB_4$, $Ti_2SnB_2$, and $Ti_3SnB_4$ (Fig. 2a) were thermodynamically stable with respect to bulk A and TiB or $Ti_3B_4$. However, further Ti-A-B ternary variable-composition searches confirmed that only $Ti_2InB_2$ is thermodynamically more stable than competing Ti-B, Ti-In and In-B binary phases in the Ti-In-B ternary system at ambient pressure, while $Ti_2SnB_2$ is stable at a high pressure of 10 GPa. Other ternary candidates suggested by the binary variable-composition search were found to fall slightly short of thermodynamic stability and could not be synthesized under pressures lower than 10 GPa.

The optimized structure of $Ti_2InB_2$ with high-precision settings is shown in Fig. 2b. The detailed structures of $Ti_2InB_2$ and $Ti_2SnB_2$ are shown in Supplementary Figs. 2 and 3, respectively. The predicted $Ti_2InB_2$ and $Ti_2SnB_2$ compounds possess typical characteristics of known MAX phases: a layered hexagonal structure (space group $P\bar{6}m2$ (No. 187), see details in Supplementary Table 1), two M (Ti) layers, and one A (In) layer close packed along an HCP A-B-A sequence. Because the B/Ti ratio (1.0) in $Ti_2InB_2$ and $Ti_2SnB_2$ is higher than those of X/M ratios (1/2, 2/3 or 3/4) in conventional MAX phases, boron atoms occupy the X sites between M layers and form a graphene-like layer (Supplementary Figs. 2 and 3) instead of a plane of equilateral triangles. The number of X (B) atoms per layer in $Ti_2InB_2$ or $Ti_2SnB_2$ is two times that for C- or N-containing MAX

phases, which implies the existence of B–B covalent bonds that could provide a stiffer structure than conventional MAX phases. Theoretical calculations (Supplementary Table 2) show that Young's modulus along the $x$- and $y$-directions, i.e., in the plane of the boron layer, is much larger than those for $Ti_2AlC$ or $Ti_3AlC_2$. Phonon stability for these two structures was also confirmed (Supplementary Fig. 4). Therefore, we conclude that a 212 B-containing MAX phase can be formed instead of the traditional form of $M_{n+1}AX_n$.

**Electronic properties of predicted $Ti_2InB_2$.** The electronic structures shown in Fig. 2c reveal that the predicted B-containing ternary compound possesses typical electronic features of MAX phases. Similar to those known MAX phases, this new B-containing MAX phase is revealed to be metallic. It can be seen that for $Ti_2InB_2$, the bonding (antibonding) states between the $d$ orbitals of Ti atoms and the $p$ orbitals of B or In atoms are located below (above) the Fermi level, whereas the non-bonding states of Ti are located between these bonding and antibonding states (near the Fermi level), and predominantly contribute to the metallic nature of $Ti_2InB_2$. This electronic structure is qualitatively similar to that for conventional MAX phases. Electron localization function (ELF) calculations showed that electron accumulation occurs between adjacent boron atoms, which reveals the 2c-2e (two center-two electron) nature of the B–B bonds (Supplementary Fig. 5) in $Ti_2InB_2$, similar to that in $AlB_2$-type compounds, from which 2D hydrogen–boron sheets have been recently obtained via cation exchange[29]. However, this situation is different from that for conventional boron clusters derived from electron-deficient multi-center 2e bonding[30], where filled octets cannot be achieved via 2c–2e bonding with only three valence electrons of boron. A Bader charge analysis showed that 0.87|e| was transferred from a Ti to a B atom, which resulted in

the formation of 2c–2e bonds between B atoms. It is noteworthy to mention that the charge separation of Ti and B and B–B 2c–2e bonds in $Ti_2InB_2$ is close to the situation in $TiB_2$ (Supplementary Fig. 5). The boron atoms in MAB phase $Fe_2AlB_2$ arrange along B–B zig-zag chains through the formation of B–B 2c–2e bonding (Supplementary Fig. 5). Similar electronic features can be found for another predicted structure, $Ti_2SnB_2$, as shown in Supplementary Fig. 6.

**Possibility of indium removal from $Ti_2InB_2$.** The interlayer A of MAX phases can be removed by etching with an appropriate acid, generally HF, which leads to the formation of a series of attractive materials, MXenes[6,31]. To evaluate the possibility of In removal from $Ti_2InB_2$, the separation energy for different interfaces along the [001] direction of $Ti_2InB_2$ was calculated (Supplementary Fig. 7). The separation energy for the Ti/In interface was found to be 3.27 J m$^{-2}$, whereas that for the Ti/B interface was 8.36 J m$^{-2}$. Therefore, the bonding between A (In) and M (Ti) is much weaker than that between M (Ti) and X (B), and is similar to that for the conventional MAX phases that can be engineered to 2D MXenes. For a clear comparison, the separation energy for the Ti/Al(001) and Ti/C(001) interfaces of $Ti_2AlC$ was calculated to be 5.66 and 11.90 J m$^{-2}$, respectively. The Ti–In bonding in the newly predicted MAX phases is much weaker than that for Ti-Al bonding in $Ti_2AlC$. The Ti/In-to-Ti/B separation energy ratio is 39%, which is also much smaller than the Ti/Al-to-Ti/C ratio of 48%. Consequently, the present calculations reveal the possibility of obtaining 2-D MXenes from $Ti_2InB_2$ by selective removal of indium through an appropriate approach. Phonon dispersion calculations (Supplementary Fig. 8) show that the hexagonal TiB structure is dynamically stable. However, our calculations show that the TiB orthorhombic structures are thermodynamically more stable (Supplementary Table 1). This suggests that the removal of indium under mild conditions (e.g., by chemical etching) may produce hexagonal TiB MXene, similar to conventional MXenes obtained by HF etching. Surface functional groups, like F, Cl, OH, and O, attributes significantly to the property modifications of conventional MXenes. We studied the electronic structures of TiBX (X = F, Cl, OH, and O) and found that a metal-to-semimetal transition appears in the functional 2D TiB (Supplementary Fig. 9). Moreover, the first-principles molecular dynamics (FPMD) simulations revealed that a phase transition from hexagonal TiB to orthorhombic TiB compounds may occur due to the strong thermal activation of lattice vibrations under high temperature conditions (Supplementary Fig. 10). The separation energy for the Ti/Ti interface of orthorhombic TiB was calculated to be 3.87 J m$^{-2}$, which is comparable with that of Ti/In interface and indicates the laminated nature of TiB (Cmcm). The orthorhombic phase (Cmcm) that exhibits layered characteristics (Supplementary Fig. 11) was experimentally obtained under high-temperature conditions and will be shown in a later section.

**Experimental synthesis and characterization of $Ti_2InB_2$.** $Ti_2InB_2$ was synthesized from a mixture of Ti, In and B powder through a solid-state reaction. The yield of $Ti_2InB_2$ was very sensitive to the experimental conditions, which were optimized after many tests (See Methods and Supplementary Figs. 12–17 and Table 3 for details). In the as-grown samples, there were many impurities, which were mainly $TiB_2$ and Ti-In phases (Supplementary Figs. 13 and 14). Adjustment of the starting compositional ratio of Ti, In and B enabled the final yield of $Ti_2InB_2$ to be maximized, while the amount of $TiB_2$ was minimized (Supplementary Fig. 15). Ti–In phase impurities can be removed by chemical etching, whereas $TiB_2$ is extremely stable

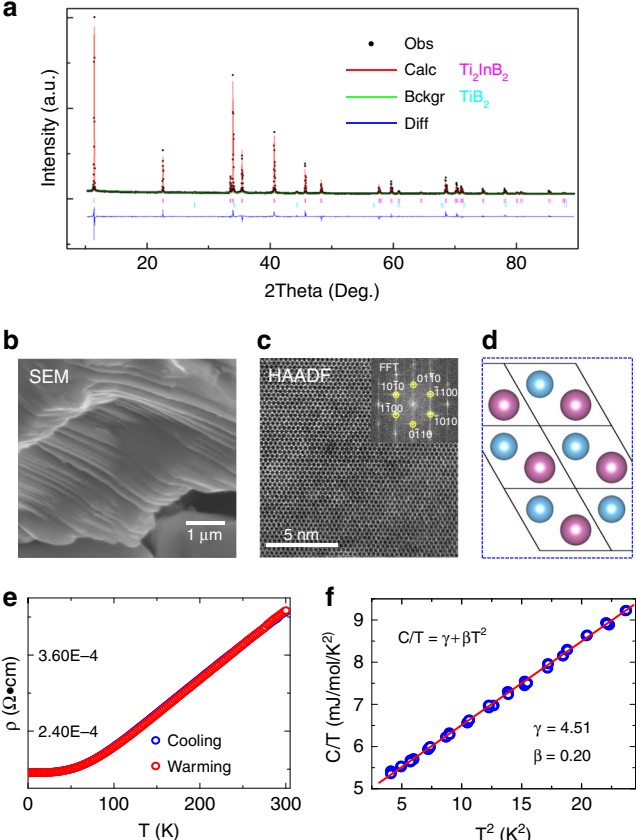

**Fig. 3** Experimental characterization of synthesized $Ti_2InB_2$ compound after HCl etching. **a** Powder XRD pattern with Rietveld analysis using GSAS package[42]. A small amount of $TiB_2$ is evident. **b** SEM image of a particle showing a laminated structure. **c** HAADF-STEM image from the [001] direction. The inset shows the corresponding FFT pattern. **d** Crystal structure viewed from the z-direction (blue and purple spheres respectively indicate Ti and In atoms). Boron is not shown and the black lines represent the unit cells. **e** Electrical resistivity. **f** Heat capacity plotted as C/T vs. T$^2$ at low temperature. The red line represents a linear fit to the data

(Supplementary Fig. 16). Therefore, after HCl acid etching, only $Ti_2InB_2$ (93.7 wt%) and a small amount of $TiB_2$ (6.3 wt%) remained, as shown in Fig. 3a, and the obtained $Ti_2InB_2$ was crystallized in the predicted structure (space group $P\bar{6}m2$). In a Rietveld analysis, the optimum fit was obtained by assuming a preferred orientation of (00l) for $Ti_2InB_2$, which suggested a material with a lamellar crystal structure. A laminar structure was clearly evident from scanning electron microscopy (SEM) observations, as shown in Fig. 3b. This is consistent with simulation results, in that the separation energy for the Ti/B interface is much larger than that for the Ti/In interface, which implies the possibility of obtaining 2D TiB sheets by the selective removal of In. The crystal structure was further confirmed by high-angle annular dark-field scanning transmission electron microscopy (HAADF-STEM) (Fig. 3c). Observation along [001] direction provides an atomic image of $Ti_2InB_2$ in the x–y plane, with hexagonal patterns recording the projection of In and Ti atoms on the plane, which is consistent with the prediction (Fig. 3d). A fast Fourier transform (FFT) of the HAADF image also indicates that $Ti_2InB_2$ crystallizes in the hexagonal structure. A homogenous composition ratio of Ti:In:B = 1.95:1:2.08 was confirmed by energy-dispersive X-ray spectroscopy (EDS) (Supplementary Fig. 17). The physical properties of $Ti_2InB_2$ were also measured and the results are shown in Fig. 3e, f. The synthesized compound

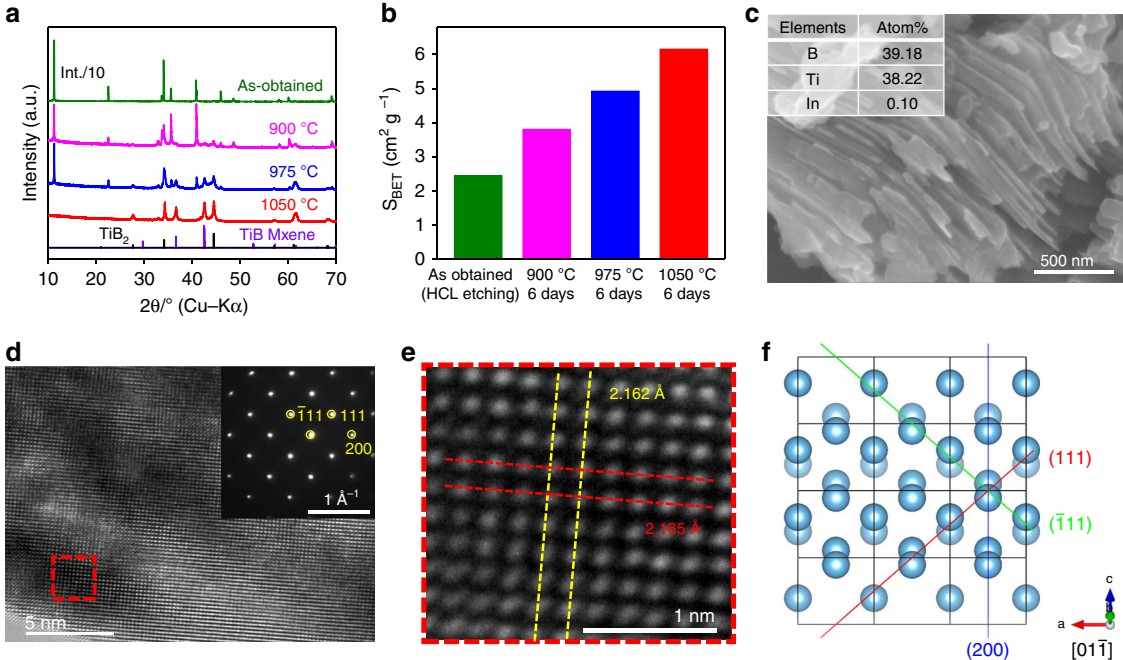

**Fig. 4** Structure characterization of layered TiB. **a** XRD patterns for samples prepared by exposure of as-obtained $Ti_2InB_2$ powder to a vacuum (about $10^{-4}$ Pa) as a function of the temperature after 6 days. **b** Corresponding specific surface area change of samples in **a**. **c** Typical SEM image of the TiB phase obtained at 1050 °C for 6 days under vacuum (about $10^{-4}$ Pa); inset shows the atomic ratio for this sample. **d** HRTEM image of the TiB phase along the [01$\bar{1}$] direction; inset shows the corresponding SAED pattern. **e** Enlarged HRTEM image from **d** showing the interlayer spacing for the (111) and ($\bar{1}$11) planes. **f** Simulated crystal structure of the TiB phase with orthorhombic group (*Cmcm*) along the [01$\bar{1}$] direction, where the blue spheres represent Ti atoms (other atoms are not shown)

exhibits metallic behavior, which is consistent with the electronic structure calculations shown in Fig. 2c. Low temperature heat capacity ($C_p$) data (Fig. 3f) were fitted well using $C_P/T = \gamma + \beta T^2$, where $\gamma$ is the coefficient of the temperature-linear $C_p$ ($\gamma T$) that describes the contribution from conduction carriers, and $\beta$ is the coefficient of the Debye $T^3$ term ($\beta T^3$) associated with propagating acoustic phonons.

**Synthesis of layered TiB**. Following the conventional etching method for producing MXenes, we first attempted to obtain 2D TiB MXene by immersion of $Ti_2InB_2$ powder in 50% HF for 12 h at room temperature. This resulted in total dissolution of the MAX phase, and even $TiB_2$ was dissolved given sufficient time (Supplementary Fig. 18). Considering the intrinsically low melting point and high vapor pressure of In metal, a dealloying strategy was adopted to exfoliate the In layers. SiC was used as an oxygen scavenger to prevent the oxidation of compounds during the removal of In at high temperature (Supplementary Fig. 19). The hypothetical dealloying reaction process is described as:

$$Ti_2InB_2 \rightarrow 2TiB + In(vap.) \qquad (1)$$

Indium atoms were gradually extracted from $Ti_2InB_2$ and coated on the inner wall of a silica glass tube outside the furnace (Supplementary Fig. 20); In was almost completely removed at 1050 °C for 6 days under vacuum conditions (about $10^{-4}$ Pa). The weight loss for the sample was in good agreement with the mass of In contained in the $Ti_2InB_2$ compound. Figure 4a and Supplementary Fig. 21 show that the main dealloyed products are TiB MX compounds with an orthorhombic structure (*Cmcm*), together with $TiB_2$ as an impurity phase in the prepared $Ti_2InB_2$.

The specific surface area increased from 2.45 to 6.16 m$^2$ g$^{-1}$ with increasing heating temperature, which also indicates that In species were gradually extracted from $Ti_2InB_2$ (Fig. 4b). Unfortunately, the lateral dimension of the parent $Ti_2InB_2$ was not kept

after In was extracted, and the obtained TiB showed smaller particle size than the parent phase but still with layered structure (SEM images in Fig. 4c), which indicates that the original laminated structure was changed during the dealloying process at high temperatures. However, the obtained TiB can be considered to be a layered material where the layers are connected by Ti-Ti metallic bonds as we discussed above. An EDS analysis (Supplementary Fig. 22) indicated the presence of mainly Ti and B with a molar ratio of ~1:1 and with almost no In residue, which is consistent with the XRD results. A high-resolution transmission electron microscopy (HRTEM) image was acquired along the [01$\bar{1}$] direction of a TiB flake, which revealed bright atomic columns arranged in a square-like pattern, directly corresponding to the stacked Ti atoms (Fig. 4d). An enlarged HRTEM image (Fig. 4e) and the corresponding selected area electron diffraction (SAED) pattern (Fig. 4d inset) of exfoliated TiB flakes indicated an interplanar spacing of 2.162 Å for the (111) plane, which is consistent with the value of 2.127 Å obtained for the simulated structure (Fig. 4f). The (021) and (200) interplanar spacings were also measured to be 2.462 and 1.588 Å, respectively, which is in agreement with the simulation results (Supplementary Fig. 23). It should be noted that high temperature often induces recrystallization, and results in the formation of non-layered, bulk 3D cubic phase by selective loss of the A element from the MAX phase[32–35]. However, local displacements of Ti and B atoms caused by the high temperature led to the generation of an orthorhombic (*Cmcm*) phase with a layered structure during removal of the In species. As shown in Supplementary Fig. 24, we consider this orthorhombic (*Cmcm*) phase to have originated from a phase change of hexagonal (*P$\bar{6}$m2*) with a similar layered structure, which can be confirmed by the simulation results of FPMD at 1273 K for 20 ps (Supplementary Fig. 10). After the FPMD simulation of 10 ps, the space group of TiB changed from *P$\bar{6}$m2* to *Cmcm* mainly due

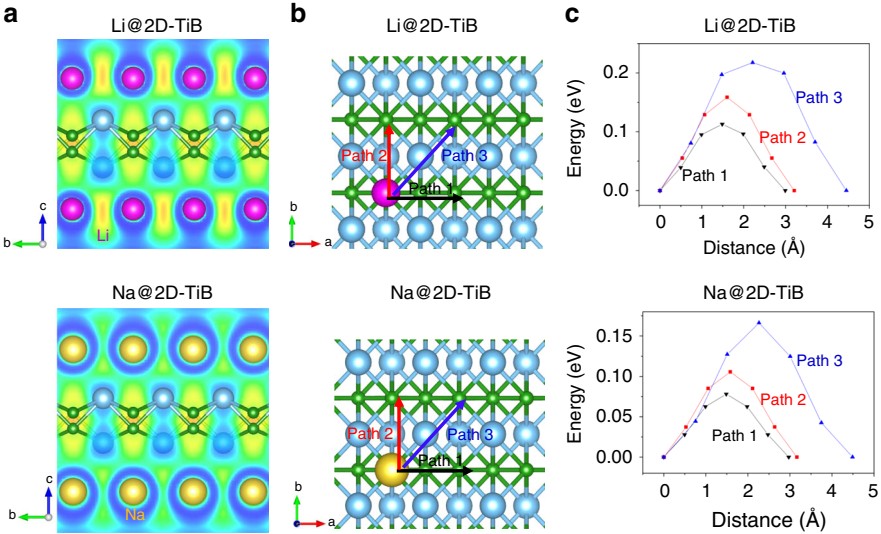

**Fig. 5** Adsorption and diffusion behavior of Li and Na on 2D TiB (*Cmcm*). **a** ELF maps of a pristine TiB monolayer with one layer of Li atoms and one layer of Na atoms. **b** Considered diffusion paths for Li and Na on the TiB monolayer. **c** Calculated diffusion energy barriers along the paths in **b**. The purple and yellow spheres represent Li and Na atoms, respectively

to interlayer slip. With further simulated heat treatment of 10 more picoseconds, the boron hexagonal rings became collapsed and boron chains instead appeared, which is the typical character of orthorhombic TiB (Supplementary Figs. 1 and 10). The real time scale of phase change from hexagonal TiB to orthorhombic phase (Cmcm) can be too long to be repeated by the FPMD simulations. However, the simulation results can be regarded as a theoretical support for the phase transition in the experiment. The phonon band structures of orthorhombic TiB structures with space groups of *Pnma* and *Cmcm* were calculated and shown in Supplementary Fig. 25. Both structures are found to be dynamically stable. Furthermore, the computed free energy (Supplementary Fig. 25) reveals that *Pnma* TiB is always more stable than the *Cmcm* phase in the full temperature range from 0 to 2000 K though the energy difference is very small (around ~0.005–0.01 kJ mol$^{-1}$). This means that the chance for the transition from *Cmcm* to *Pnma* can be little due to the weak thermodynamic driving force, which explains the experimental result that the metastable *Cmcm* phase is the dominant product by heat treatment at 1050 °C.

**Prospect of layered TiB as anode material for ion batteries.** Although impurities, such as TiB$_2$ and 3D TiB (*Pnma*), were present, a new layered material, TiB (*Cmcm*), was successfully prepared by dealloying the new Ti$_2$InB$_2$ MAX phase. Although the layered TiB (*Cmcm*) is not comparable with 2D MXenes, the successful removal of In atoms suggests the possibility to obtain TiB MXene at mild conditions. Therefore, it is necessary to investigate the potential applications of 2D TiB in advance. To evaluate the thermal stability of the 2D TiB structure at elevated temperatures for applications, we carried out FPMD simulations at 1273 and 1773 K for 10 ps (Supplementary Fig. 26). No sign of disruption or structural decomposition even in the 1773 K simulation. Therefore, it is expected that 2D TiB will be stable at temperatures as high as 1773 K for practical applications. The intercalation of Li$^+$ or Na$^+$ ions between MXene sheets makes them promising materials for Li- or Na-ion batteries. The charge/discharge reaction that should occur at the anode is: TiBLi(Na) ↔ TiB + Li$^+$(Na$^+$) + e$^-$. To confirm the potential of layered TiB as an anode material for Li/Na ion batteries, adsorption of Li and Na atoms on the surface of a TiB monolayer was first studied using

DFT calculations. These calculations revealed that each TiB primitive cell (Ti$_2$B$_2$) can accommodate up to two Li or two Na atoms (the adsorption of extra Li/Na atom would be thermodynamically unfavored.), which corresponds to a composition of TiBLi or TiBNa (Fig. 5a). Therefore, the theoretical specific capacity of TiB for Li or Na ions was calculated to be 480 mAh g$^{-1}$, which is significantly higher than those of the conventional MXenes Ti$_3$C$_2$ (320 mAh/g for Li$^+$)[11], Ti$_2$C (359 mAh g$^{-1}$ for Na$^+$)[36] and even higher than that of the commercial anode material, graphite (372 mAh g$^{-1}$ for Li$^+$)[11]. The open circuit voltage (OCV) for the intercalation reactions involving Li$^+$ and Na$^+$ ions on a TiB surface was respectively estimated to be as low as 0.33 and 0.17 V, respectively, which are both much smaller than that for Ti$_3$C$_2$ (0.62 V for Li$^+$) and comparable with that for graphite (about 0.2 V for Li$^+$) and the hypothetical TiC$_3$ (0.18 V for Na$^+$)[11,36]. The calculated ELF across the absorbed Li/Na atoms and TiB monolayer show that the electron cloud spread out in the metal layers can screen the repulsion between the positive metal ions (Fig. 5a). A Bader charge analysis revealed that the charge transfer from absorbed Li atoms to the TiB monolayer was 0.80 | e|/Li, while one Na atom lost 0.48 |e| upon adsorption. Therefore, Li/Na atoms can be stabilized by the coulombic attraction between negatively charged B and positively charged Li/Na atoms.

The diffusion energy barrier for Li or Na on the TiB surface is critical for determining the charge−discharge rate in Li/Na ion batteries. The diffusion energy barrier along three different pathways (Fig. 5b) between the most stable nearest-neighboring adsorption sites of Li and Na on a $3 \times 3$ TiB supercell was calculated using the climbing-image nudged elastic band (CI-NEB) method[37]. Figure 5b, c show that Li ions move along Path 1 and 2 with lower energy barriers of 0.11 and 0.16 eV, while Path 3 has the highest energy barrier of 0.22 eV. Similarly, for Na diffusion on the TiB surface, the energy barriers along Paths 1, 2, and 3 are respectively 0.08, 0.11, and 0.17 eV. Therefore, the new TiB monolayer should have a high charge-discharge rate due to these computed low energy barriers for Li and Na diffusion, which are very competitive with those for conventional MXenes (e.g. 0.18 eV for Li diffusion on Ti$_3$C$_2$), graphite (0.30 eV for Li diffusion), anatase TiO$_2$ (0.35–0.65 eV for Li diffusion) and the predicted TiC$_3$ (0.18 eV for Na diffusion)[11,36]. We noted that the diffusion activation energy for Li$^+$ and Na$^+$ ions on oxidized TiB surface gets increased (0.31~0.68 eV for Li$^+$ and 0.22~0.50 eV for

$Na^+$) but is still competitive to the reported conventional materials (Supplementary Fig. 27). Considering the high specific capacity, low OCV and low energy barriers for $Li^+$ and $Na^+$ ions, the 2D- TiB (*Cmcm*) could be a promising alternative material to the commercial graphite anode in Li/Na ion batteries. Moreover, the diffusion of $Li^+/Na^+$ ions on hexagonal TiB surface was investigated (Supplementary Fig. 27), because the present research indicates the possibility to obtain TiB MXene at mild conditions. It shows that the diffusion energy barriers of $Li^+/Na^+$ ions on clean hexagonal TiB surface can be as low as around 0.02 eV. On the oxidized surfaces, the calculated energy barriers for $Li^+$ and $Na^+$ are, respectively, 0.23 and 0.19 eV.

## Discussion

A progressive calculation-based approach was employed to predict and synthesize the boron-containing MAX phase $Ti_2InB_2$. Application of the proposed approach effectively screens out those material systems with a low possibility to produce new MAX phases and involves a much lower computational cost than direct trial and error or high-throughput calculations for ternary compound systems. The obtained $Ti_2InB_2$ compound possesses typical features of MAX phases, such as a layered hexagonal structure (symmetry *P6m2*) and a close-packed atom stacking sequence, the co-existence of weak Ti-In metallic and strong Ti-B covalent bonding along the [001] direction, metallic electrical resistivity and ceramic-like low heat capacity. With solid evidence from theoretical calculations and experimental validation, we have provided a clear example to extend the MAX phase family to boron-containing systems. Moreover, DFT calculations revealed that the separation energy for the Ti/In metallic interface is much smaller than that for the Ti/B covalent interface, which indicates the possibility of In atom removal while maintaining the TiB layered structure. Following this theoretical prediction, a layered TiB structure (*Cmcm*) was obtained by dealloying the $Ti_2InB_2$ MAX phase under vacuum conditions. The obtained TiB exhibited superior stability over the conventional MXenes and maintained its layered structure at high temperature. DFT calculations also predicted that the conductive TiB would be a promising anode material for Li/Na ion batteries with respect to a low OCV, a low Li/Na ion diffusion barrier, and a high theoretical Li/Na ion capacity. The present research will extend the fascinating class of MAX phases and MXenes.

## Methods

**High-throughput structure search**. Structure searches were conducted using a combination of Universal Structure Predictor: Evolutionary Xtallography (USPEX)[25–28] and the Vienna Ab initio simulation package (VASP)[38,39]. This approach was used to identify the structure with the lowest free energy for a given composition or in a composition range under given external conditions (0 K and 1 atm). Three types of structure searches were performed: fixed, binary variable composition, and ternary variable. For a fixed composition structure search, the calculation scheme comprised two steps: global optimization and local optimization. Structure initiation and variation were performed using USPEX in the global optimization step. For a variable composition structure search, two or three elements/compounds were adopted as reference components in a search of the target system. All possible combinations of the reference components were considered in the USPEX structure search, limited only by the number of atoms number per unit cell.

**Density functional theory calculations**. In the evolutionary structure search, thousands of structures were relaxed using DFT calculations. Therefore, sufficiently good settings were used to reduce the calculation cost: the energy cutoff for the plane-wave basis set expansion was set at 400 eV, and the Monkhorst–Pack[40] k-point mesh solution in reciprocal space was $2\pi \times 0.06$ Å$^{-1}$ for all structures. For structure refinement, the cutoff energy and k-point were, respectively, improved to 600 eV and $2\pi \times 0.04$ Å$^{-1}$ to optimize the most stable structures obtained in the evolutionary structure searches. The atomic positions and unit cell volumes were fully relaxed until the total energy converged within 0.001 eV. These calculation settings were also adopted for the electronic structure calculations. Specifically, a denser k-mesh of $2\pi \times 0.02$ Å$^{-1}$ was used for the density of states (DOS) for the

predicted structures. Phonon dispersion calculations were performed to confirm the dynamic stability of all thermodynamically stable structures using VASP[38,39] and Phonopy[41]. We performed first-principles molecular dynamics (FPMD) calculations using the VASP[38,39] to simulate the transition from hexagonal TiB to orthorhombic phase and confirmed the thermal stability of 2D-TiB at elevated temperature. The FPMD simulations were performed with an NVT ensemble (keeping the number of atoms *N*, volume *V* and temperature *T* constant) with a Nosé thermostat. The time step of 2.0 fs used for each simulation. To compare their thermodynamic stabilities of different phases of TiB at different temperature, their Helmholtz free energies were calculated by following equation:

$$F(T) = E_{tot} - T \cdot S(T) \qquad (2)$$

The total energy $E_{tot}$ was calculated by VASP[38,39], the entropy S(T) was obtained by VASP and Phonopy[41] calculations.

**Synthesis of predicted $Ti_2InB_2$**. $Ti_2InB_2$ was synthesized from Ti (99.9%), In (99.99%), and B (99%) powder using a solid-state reaction route. The chemicals were purchased from Kojundo Chemical Laboratory Co., Ltd. Single-phase $Ti_2InB_2$ could not be directly obtained, because it competed with other stable phases (mainly $TiB_2$ and Ti-In phases). Finally, the yield was maximized by varying many different experimental conditions: the optimized values were 1100 °C for 36 h without further annealing using a quartz tube sealed with Ar gas and Mo foil as the protector, and excess amounts of both Ti and In (>25%). The samples used for characterization and property measurements were prepared with increased amounts of Ti and In by 50%. More information regarding the optimization process of temperature, the type of crucible, the atmosphere, the annealing process and the initial composition ratio is provided in Supplementary Figs. 12–17 and Table 3. After the solid-state reaction, chemical etching was applied to remove the impurity phases. Around 2 g samples were immersed in diluted hydrochloric acid (2 mol L$^{-1}$, 100 mL) at room temperature for 10 h. Ti–In phases ($Ti_3In$, $Ti_3In_4$, $Ti_{2.2}In_{1.8}$) were gradually dissolved with the evolution of $H_2$ bubbles, and the color of the solution became purple ($TiCl_3$). $TiB_2$ exhibited high stability against etching, and could not be removed (Supplementary Fig. 16). Nitric acid was also used for etching; however, $TiO_x$ was produced as a new impurity.

**Synthesis of layered material TiB**. The as-prepared $Ti_2InB_2$ powders were filled into a Mo tube (in-house built with Mo foil, Nilaco Corporation, >99.95%). Several SiC (Kanthal Corporation) ingots were placed separately. The molybdenum tube and SiC ingots were heated at 900–1050 °C (5 °C min$^{-1}$) for 6 days in a dynamically evacuated quartz tube (about 10$^{-4}$ Pa) as shown in Supplementary Fig. 20. This heating step was effective for improving the dealloying of $Ti_2InB_2$, and the yield of TiB gradually increased with increasing annealing temperature. Finally, the optimized condition for the dealloying reaction was determined to be 1050 °C for 6 days. The obtained product TiB was a puce color, different from the dark gray color of the parent $Ti_2InB_2$.

**Characterization of $Ti_2InB_2$ and TiB**. Powder X-ray diffraction (XRD) measurements were performed using an X-ray powder diffractometer with Cu Kα radiation (Bruker, D8 Advance). The morphology of the sample was evaluated using field-emission scanning electron microscopy (FE-SEM; JEOL, JSM-7600F) and the component elements were analyzed using energy-dispersive X-ray spectroscopy (EDS; JEOL, JED-2300). Transmission electron microscopy (TEM) and high-angle annular dark-field scanning transmission electron microscopy (HAADF-STEM) images were obtained using a JEOL JEM-ARM200F atomic resolution analytical electron microscope operated at an accelerating voltage of 200 kV. Resistivity and heat capacity were measured with a physical properties measurement system (PPMS, Quantum Design). The Brunauer−Emmett−Teller (BET) specific surface areas of the samples were determined from nitrogen adsorption–desorption isotherms measured at −196 °C using an automatic gas-adsorption instrument (BELSORP-mini II, MicrotracBEL).

## Data availability

All data supporting the findings of this study are available within the article and the Supplementary Information file, or are available from the corresponding authors upon reasonable request.

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

## Acknowledgements

This work was supported by the Ministry of Education, Culture, Sports, Science and Technology (MEXT) of Japan through the Element Strategy Initiative to Form Core Research Center. This work was supported in part by the ACCEL program sponsored by the Japan Science and Technology Agency (JST). This work is supported by National Natural Science Foundation of China (No. 51872242). T.-N.Y. was supported by a JSPS fellowship for International Research Fellows (No. P18361). H.H. was supported by the Japan Society for the Promotion of Science (JSPS) through a Grant-in-Aid for Scientific Research (S) (No. 17H06153). We thank M. Ichihara and M. Sasase (Tokyo Institute of Technology) for technical support with the TEM measurements. J.Wu thanks Y. Muraba (Tokyo Institute of Technology) for his help on sample synthesis. All of the calculations were performed on the supercomputer at the National Institute for Materials Science (NIMS), Tsukuba, Japan.

## Author contributions

H.H., J.W., and T.T. proposed the idea behind the research. H.H. supervised the research. J.W. conducted high-throughput structure prediction and density functional theory calculations. Y.G. performed the preliminary synthesis of Ti$_2$InB$_2$. J.Wu. and T.-N.Y. performed the synthesis and characterization of Ti$_2$InB$_2$ and TiB. N.M. performed FPMD simulations and free energy calculation for TiB. J.W., T.-N.Y., J.Wu., and H.H. co-wrote the paper. All authors discussed the results and commented on the manuscript.

## Additional information

**Competing interests:** The authors declare no competing interests.

