## [Peer Review File · Nature Communications]

Reviewers' comments:

Reviewer #1 (Remarks to the Author):

I like this paper a lot, but several things need to be fixed and clarified:

1. Abstract mentions hexagonal 2D-phase of TiB as "confirmed" by synthesis, but the main text of the article seems to suggest that only the bulk phase was hexagonal and that at high temperatures at which In was removed and the 2D-material formed, the orthorhombic phase was formed. In this case, what kind of confirmation was given to hexagonal TiB? If so, can we call this material a MXene or MXene-like?
2. Abstract mentions space groups being "similar", but such notion doesn't exist. One can speak of group-subgroup relations, but not of similarity of space groups.

Everything else, including structure prediction, NEB calculations, estimates of capacity, is perfect.

Reviewer #2 (Remarks to the Author):

MAX phases have recently drawn great attentions due to their unique structure and properties. On the other hand, two-dimensional (2D) layered materials can be directly obtaining from the MAX phases through solution etching method, providing another roadmap for synthesizing 2D materials. Recent years, theoretical calculations play the key role in accelerating the discovery of new materials and understanding the physical mechanism behind experimental phenomenon. In this work, the authors predicted a novel MAX phase of Ti_2InB_2 . Then, they successfully synthesized it and got a 2D layered material of TiB. The performance of TiB as anode material for lithium-ion battery or sodium-ion battery has also been explored. Overall, the results are interesting and the finding material has the potential applications. But, there is a few places need to be completed as shown below.

1. The temperature-dependent stability is very important for practical applications, especially for TiB 2D materials. The authors can be explored this by performing molecular dynamical simulations.
2. How is the volume change of TiB upon adsorbing Li or Na? Whether TiB can adsorb two layers of lithium.
3. The Ti atoms in 2D TiB expose the surface, what is the degree of interaction between the layers?
4. MXene-like materials usually need surface functionalization. In view the structural character of 2D TiB, I encourage the authors to explore its surface functionalization, which might produce some interesting results.

Reviewer #3 (Remarks to the Author):

Comments to Ti_2InB_2

In general I'm impressed by the findings of the authors. The expectation based on theoretical calculations for the existence of a compound Ti_2InB_2 and the subsequent experimental proof (if this is the true story) is an excellent proof of the fruitful interaction between theory and experiment. Furthermore, the fundamental importance of Ti_2InB_2 for the discussion of MAX-phases dives a good reason for the publication in Nature.

Nevertheless, I have some critical points which should be clarified or changed before publication. The first point is the claimed analogy to MXenes. I don't see the 2D character of the TiB structure (Cmcm). Experimental access by thermal treatment up to 1000°C, SEM pictures and XRD patterns give no proof of this interpretation. The second point is the classification of hexagonal TiB as metastable and the focusing on TiB (Cmcm) as the only representative while ignoring the experimentally proven TiB (Pnma).

As a non-native speaker I give no comments on language, grammar, spelling,... (and apologize for my own mistakes)

Further comments

P1, abstract, line 5 expression „close“: The similarity to MAX phases (I agree) results not from a symmetry relation (although it can be discussed if one refers to the similarity and differences between octahedral surroundings and trigonal prisms). So I recommend another expression.

P2, Discussion on TiB/Ti₃B₄: see comments to Figure S1 (s. below).

Line 6: “3” must be a subscript, please correct it within the whole manuscript and supporting information.

P2, second paragraph, discussion on MAX-/MAB-phases: Just to focus on the orthorhombic metrics of the MAB phases is not sufficient for the distinction to the MAX phases. In MAB phases the topology gives 3 different characters for the connection of the trigonal prisms (in agreement to the orthorhombic structure) following the B-B-bonds, connection of the trigonal planes and perpendicular to the Al-layers. In the case of the MAX phases, the equivalence of a and b by symmetry is indeed a consequence of the symmetry.

P2, third paragraph, classification of TiB and Ti₃B₄ as layer structures: All investigations on these type of compounds (for example Acta Cryst. B, 2015, 71, 777) show that there is a significant interaction between the transition metal atoms (also mentioned by the authors) and additionally via the seventh TM-B bond (for example in CrB: 4x2.19 Å, 2x2.22Å, 1x2.29 Å) This is in contrast to the MAX-phases, where the interaction is much more anisotropic.

P5, line 3: The reference to “stable” TiB and Ti₃B₄” is not clear. Which modification of is meant? Pnma (existent) Cmcm (not yet found in the binary system. Why is no reference to existing and stable TiB₂, for example 2TiB + TiB₂ = Ti₃B₄ (or Ti₂B₃ with the structure of V₂B₃, or Ti₅B₆?)

P5, second paragraph: Comment of a boron chemist: The structure chemistry of transition metal borides is dominated by trigonal prisms. Octahedral surroundings as they occur in MAX phases are extremely(!!) rare. So a B-containing MAX—phase would be very surprising (comment: hexagonal borides like Ti₅Si₃B are not comparable, because the boron content is unknown, Ti₅Si₃B_x (x ~ 0.5??).

P5, last paragraph, discussion on ELF and Bader charge. I agree to the results on charge separation in Ti₂InB₂. But the similarity to MAB phases and also binary transition metal borides requires, that results on these compounds are used for comparison (and cited) and not the results on boron-rich compounds.

P6, Discussion on the stability of hexagonal TiB and orthorhombic TiB (Cmcm). I definitely disagree to describe a transition of a “delaminated or dealloyed” Ti₂InB₂, i. e. hypothetical hexagonal TiB, to TiB (Cmcm, i.e. CrB-type) as a lattice relaxation. There is a completely different structure with different bonding characteristics, that can't be achieved by a lattice relaxation (I understand this term , that you have a change of the lattice parameter by some extent (15?) and therefore maybe by a change of the crystal system).

P7, line 6, expression “6-member rings”: The structure of Ti₂InB₂ contains no such rings of In and Ti atoms. The motif of Ti atoms and In atoms as well are from a closest packing (topologically a 36-net). The boron atoms form 6-member rings (63-net). I agree, that the projection of both atoms form hexagons.

P8, figure 3d: who is who? (but it doesn't matter, it's the same motif).

P8/P9: discussion on the synthesis of quasi-2D MXene-like TiB: For this paragraph I have two questions. First, I do not understand the role of SiC. SiC is not known for its high affinity vs. oxygen. The surface of “several SiC ingots” seems not to be sufficient (in my opinion, but I haven't seen the

samples). I'd suppose, that Mo has a higher affinity to O_2 , especially, as MoO_3 is quite volatile. Furthermore, I don't know what the source of oxygen can be. So is there any proof, that SiC acts as a oxygen scavenger? For example, Zr sponge is really an oxygen scavenger (if this is an issue for the formation of TiB from Ti_2InB_2 by "dealloying"). Concerning the "dealloying" of In. Is there any experimental proof for the reaction? The second point is much more important for the interpretation of the results. I interpret the results as a decomposition of Ti_2InB_2 into metastable(?) TiB (CrB-type) release of In (evaporation? reaction?...). I don't agree, that a 2D-MXene-like product is formed. The P-XRD-pattern "as obtained" has more or less the same number of reflections with comparable line width as the other XRD patterns. For a real 2D structure like the defoliated MXenes I expect a XRD pattern with a strong emphasis of the reflections of the 2D layer (for example 00l for hexagonal TiB) and a significant broadening. This is not the case. So I see the formation of CrB-type TiB, that is quite well crystallized, although the particle size is smaller than the originally ones (as expected according to the formation from a decomposition).

P9, figure 4a and b: Fig 4a shows the formation of TiB between 975 and 1015°C. This increase is not related to a significant increase of the specific surface (4b). Therefore, I'd interpret these observations that the decomposition of Ti_2InB_2 occurs with a release of In and a quite fast formation of TiB (Cmcm) without intermediate hexagonal TiB. TiB (Cmcm) behaves not like a 2D material and is quite well crystallized (line width comparable to TiB₂). Therefore the "dealloyed" (in my view decomposed) Ti_2InB_2 can't be compared to a MXene. Please keep in mind that crystals of the CrB-type or ternary borides containing the structural unit of CrB like MoAlB or Cr_2AlB_2 do not form very thin platelets like MAX-phases do. Furthermore, if one compares the SEM pictures of MXenes and figure 4c there is a significant difference. Obviously the transformation of the quite instable hexagonal TiB (no experimental proof for the occurrence as a metastable phase in the course of the decomposition despite the calculations) to TiB (Cmcm) results in microcrystalline samples of metastable (?) TiB. P9, figure 4c: A comparison to Figure S19 shows, that significant amounts of carbon and oxygen represent the missing 20%. Therefore, the interpretation (or suggestion) of the analytical results as "single phase TiB" is too optimistic and requires a further explanation.

I agree that Ti_2InB_2 represents a remarkable extension of the MAX phases by combination of the typical features of a transition metal boride (hexagonal TiB₂) with an intermetallic part (TiIn) following the principle of the MAX-phases and not the MAB-phases. The 2D character of Ti_2InB_2 is comparable to the MAX phases. I don't follow the authors in their conclusions the "dealloying" by thermal treatment at 975-1050°C results in a 2D structure of TiB (Cmcm) comparable to the MXenes. Therefore, the analogy concerning the application for example use as an anode material is not obvious as long as there is no experimental proof.

Supplementary information:

Please change the space group symbol of "P-6m2" the letter "M" to a "m".

Figure S1: In my opinion it is misleading to discuss only TiB (Cmcm) as a result of the calculations (self-fulfilling prophecy). There is no comment on the stability of experimentally established TiB (Pnma) (Acta Cryst 1954, 7, 77; Monatsh. Chem. 1960, 91, 608;) Furthermore, I miss a reference to the experimentally established compound Ti₃B₄. Both aspects should be considered for the discussion on P2 and the introduction of the manuscript. Comment from an experimentalist: The existence of a compound is not proven by calculation of a local minimum in the energy hypersurface and vice versa its non-existence by the absence of this minimum.

Table S1: Position/coordinates/site of B in the models of space group P-6m2: In my version of the International Tables for Crystallography the site 0,0,0.5 (Wyckoff 1b) is a onefold site. Therefore, the compositions given as TiB or Ti_2InB_2 should be Ti_2B and Ti_2InB , respectively. If the authors want to describe the structure model as discussed in the manuscript they have to add a second boron site (I suggest Wyckoff site 1f $2/3, 1/3, 1/2$). For the case, that there is really only one site for boron the complete discussion including the calculations of the stability must be completely changed (note: XRD-patterns will not be affected).

Reviewer #1

(1) Abstract mentions hexagonal 2D-phase of TiB as "confirmed" by synthesis, but the main text of the article seems to suggest that only the bulk phase was hexagonal and that at high temperatures at which In was removed and the 2D-material formed, the orthorhombic phase was formed. In this case, what kind of confirmation was given to hexagonal TiB? If so, can we call this material a MXene or MXene-like?

Response: We thank the reviewer for this constructive comment. Our DFT calculations indicates the possibility of In atom removal from Ti_2InB_2 while maintaining the TiB layered structure because the separation energy for the Ti/In metallic interface is much smaller than that for the Ti/B covalent interface. The high temperature treatment of Ti_2InB_2 proved that the In atoms can be removed while maintaining the layered TiB structure. However, the symmetry of TiB changed from hexagonal space group to orthorhombic one as pointed out by the reviewer. Therefore, we agree that the term "MXene-like" is not precise enough to describe the obtained TiB. Consequently, we revised the description through the manuscript to change "MXene-like TiB" to "layered TiB".

(2) Abstract mentions space groups being "similar", but such notion doesn't exist. One can speak of group-subgroup relations, but not of similarity of space groups.

Response: We thank the reviewer for pointing out this problem. We agree that the description of "similar" is not precise enough. The sentence in the abstract is revised as "...its space group was confirmed as $P\bar{6}m2$ (No. 187), which is in fact a hexagonal subgroup of $P6_3/mmc$ (No. 194), ...".

Reviewer #2

(1) The temperature-dependent stability is very important for practical applications, especially for TiB 2D materials. The authors can be explored this by performing molecular dynamical simulations.

Response: We thank the reviewer for this good suggestion. To evaluate the thermal stability of 2D TiB at elevated temperature, we carried out first-principles molecular dynamics (FPMD) simulations at 1273 and 1773K for 10 ps. There is no sign of disruption even in the 1773K simulation. Therefore, it is expected that the 2D TiB will be stable at temperature as high as 1773K. The description of the FPMD simulations were added in the revised manuscript. Figure S24 was added in the supporting information. Several sentences were added in the first paragraph of Sec. Prospect of 2D TiB as anode material for Li/Na ion batteries: "To evaluate the thermal stability of the 2D TiB structure at elevated temperatures for applications, we carried out FPMD simulations at 1273, and 1773 K for 10 ps ((Supplementary Fig. 26)). No sign of disruption or structural decomposition was seen even in the 1773 K simulation. Therefore, it is expected that 2D TiB will be stable at temperatures as high as 1773 K for practical applications." (from line 292 of page 11). A brief description of the FPMD

setting is inserted in Methods section (from line 5 of page 14).

(2) *How is the volume change of TiB upon adsorbing Li or Na? Whether TiB can adsorb two layers of lithium.*

Response: We thank the reviewer for this comment. Since the TiB used in adsorption calculations is just a monolayer, it is more convenient to consider the surface expansion/contraction upon adsorptions than volume change. Upon the adsorption of Li or Na, the surface area of TiB monolayer gets increased 2.66% or 4.46%. Based on calculated adsorption energy, TiB can only absorb one layer lithium per side. We have revised the description in the revised manuscript as “**These calculations revealed that each TiB primitive cell (Ti_2B_2) can accommodate up to two Li or two Na atoms (the adsorption of extra Li/Na atom would be thermodynamically unfavored.), which corresponds to a composition of $TiBLi$ or $TiBNa$ (Fig. 5a).**” (from line 300 of page 11)

(3) *The Ti atoms in 2D TiB expose the surface, what is the degree of interaction between the layers?*

Response: The separation energy for the Ti/Ti interface in TiB compound was calculated to be 3.87 J/m^2 , which is comparable with that of Ti/In interface in Ti_2InB_2 . Therefore, metallic bonding between the TiB layers can be expected. We added the results in Supplementary Fig. 7 and one sentence in the last paragraph of Sec. *Possibility of indium removal from Ti_2InB_2* : “**The separation energy for the Ti/Ti interface of orthorhombic TiB was calculated to be 3.87 J/m^2 , which is comparable with that of Ti/In interface and indicates the laminated nature of $TiB(Cmcm)$.**” (from line 183 of page 7)

(4) *MXene-like materials usually need surface functionalization. In view the structural character of 2D TiB, I encourage the authors to explore its surface functionalization, which might produce some interesting results.*

Response: We thank the reviewer for this very useful suggestion. We presented the geometrical structures and electronic structure of 2D TiB with different functional groups of F, Cl, OH and O in the revised manuscript. Figure S9 was added in the Supplementary Information. One sentence was added in the Sec. *Possibility of indium removal from Ti_2InB_2* : “**Surface functional groups, like F, Cl, OH and O, attribute significantly to the property modifications of conventional MXenes. We studied the electronic structures of $TiBX$ ($X=F, Cl, OH, \text{ and } O$) and found that a metal-to-semimetal transition appears in the functional 2D TiB (Supplementary Fig. 9).**” (from line 177 of page 7) The study about the diffusions of Li and Na on 2D-TiBO ($Cmcm$) and on the hexagonal 2D-TiB and TiBO were added in the Sec. *Prospect of 2D TiB as anode material for Li/Na ion batteries*. Figure S27 was added in the supporting information to show the diffusion behavior of Li and Na on new monolayers. Several sentences were added in the Sec. *Prospect of layered TiB as anode material for Li/Na ion batteries*: “**We noted that the diffusion activation energy for Li^+ and Na^+ ions on oxidized TiB surface gets increased ($0.31\sim 0.68 \text{ eV}$ for Li^+ and $0.22\sim 0.50 \text{ eV}$ for Na^+) but is still competitive to the reported conventional materials (Supplementary Fig. 27).**” (from line

325 of page 12) “Moreover, the diffusion of Li⁺/Na⁺ ions on hexagonal TiB surface was investigated (Supplementary Fig. 27) because the present research indicates the possibility to obtain TiB MXene at mild conditions. The result shows that the diffusion energy barriers of Li⁺/Na⁺ ions on clean hexagonal TiB surface can be as low as around 0.02 eV. On the oxidized surfaces, the calculated energy barriers for Li⁺ and Na⁺ are respectively 0.23 and 0.19 eV.”. (from line 330 of page 12)

Reviewer #3

Comments to Ti₂InB₂

(1) *In general, I'm impressed by the findings of the authors. The expectation based on theoretical calculations for the existence of a compound Ti₂InB₂ and the subsequent experimental proof (if this is the true story) is an excellent proof of the fruitful interaction between theory and experiment. Furthermore, the fundamental importance of Ti₂InB₂ for the discussion of MAX-phases gives a good reason for the publication in Nature.*

Response: First, we would like to thank the referee for reviewing our manuscript with a great patience and providing extremely useful comments. Based on the following point-by-point response to your comments, we hope that through the revision process we have arrived at an improved paper addressing your scientific concerns.

(2) *Nevertheless, I have some critical points which should be clarified or changed before publication. The first point is the claimed analogy to MXenes. I don't see the 2D character of the TiB structure (Cmcm). Experimental access by thermal treatment up to 1000°C, SEM pictures and XRD patterns give no proof of this interpretation.*

Response: This is a good comment. As the reviewer pointed out, the present obtained TiB is indeed not a MXene. Therefore we used the term “MXene-like TiB” in our original submitted manuscript. Because we realized now that this word still looks too strong to describe our materials, we changed the term as “layered TiB” through the whole revised manuscript, as the response to a comment by Reviewer 1. However we truly believe that the experimental obtained TiB (Cmcm) shows a quasi-2D character, as the referee can refer to Fig. S1. There is a relatively large space between Ti/Ti interface in comparison to the short and strong Ti-B bond forming a TiB layer for the Cmcm TiB, while for Pnma TiB, it is a real 3D network connected by Ti-B bonds. In order to prove the quasi-2D character of the Cmcm TiB, we have performed additional calculations. The separation energy for the Ti/Ti interface was calculated to be 3.87 J/m², which is comparable with that of Ti/In interface in Ti₂InB₂, but much weaker than that of Ti/B interface, indicating the laminated nature of the Cmcm TiB.

In addition, we used heat treatment up to 1050 °C, which is an extremely high temperature, to obtain the Cmcm TiB. This phase was mainly confirmed by our XRD, TEM and SEM-EDS measurements, which show compatible results with our theoretical predictions. Although the SEM images have already indicated that the obtained TiB showed a laminated morphology (Fig. 4c), we cannot say that we obtained

comprehensive experimental support for its 2D character, as the referee pointed out. This is mainly due to our high-temperature treatment, which may change the laminated structure of Ti_2InB_2 (Fig. 3b) and created much smaller particles. It is noteworthy that the present strategy is only our initial attempt, which only succeeds partially but provides the possibility towards the real TiB MXene. In contrast to the conventional MXenes, which totally become cubic 3D-network structures at high temperatures (*J. Am. Ceram. Soc.* **94**, 4556 (2011); *J. Electrochem. Soc.* **146**, 3919 (1999); *Mater. Sci. Eng. A* **298**, 174 (2001); *Acta Mater.* **55**, 1479 (2007)), *Cmcm* TiB can still maintain its 2D character (though was also largely modified) even at 1050 °C. To improve the description of our discovery, we revised the sentence in our manuscript as “Unfortunately, the lateral dimension of the parent Ti_2InB_2 was not kept after In was extracted, and the obtained TiB showed smaller particle size than the parent phase but still with layered structure (SEM images in Figure 4c), which indicates that the original laminated structure was changed during the dealloying process at high temperatures.”. (from line 241 of page 9)

(3) *The second point is the classification of hexagonal TiB as metastable and the focusing on TiB (Cmcm) as the only representative while ignoring the experimentally proven TiB (Pnma).*

Response: We thank the reviewer for raising this interesting discussion. We didn't actually ignore the experimental proven TiB (*Pnma*), as the referee can refer to Fig. S23, S25, and Table S1. However, we understand that our investigation seems to be not sufficient on this point. So we have made further calculations and added further information as follows.

First, we calculated the phonon band structures and Gibbs free energies of TiB with different space groups. We confirmed that both structures are dynamically stable and energy difference between these two phases are very small (around 0.005~0.01 kJ/mol). Figure S24 was added into the revised Supporting Information and one paragraph was inserted in the Sec. **Synthesis of layered TiB**: “The phonon band structures of orthorhombic TiB structures with space groups of *Pnma* and *Cmcm* were calculated and shown in Supplementary Fig. 24. Both structures are found to be dynamically stable. Furthermore, the computed free energies (Supplementary Fig. 24) show that *Pnma* TiB is always more stable than the *Cmcm* phase in the full temperature range from 0K to 2000K though the energy difference is very small (around 0.005~0.01 kJ/mol). This means that the chance for the transition from *Cmcm* to *Pnma* can be little due to the weak thermodynamic driving force, which explains the experimental result that the metastable *Cmcm* phase is the dominant product by heat treatment at 1050 °C.” (from line 269 of page 10). A brief description of the free energy calculation is inserted in Methods section (from line 10 of page 14).

It is noted that the obtained TiB (*Cmcm*), although metastable (the stability is very close to *Pnma* TiB), was confirmed by our XRD, TEM and SEM-EDS measurements, which show consistent results with our theoretical predictions. We would like to emphasize that the hexagonal TiB may only exist (with surface functional groups) at mild conditions, similar to conventional MXenes, while at high temperatures, it would

evolve to another more energy-favored structures (either *Cmcm* TiB or *Pnma* TiB). Because hexagonal TiB and *Cmcm* TiB have structure and symmetry similarities (both of them have 2D characters with a space separating the Ti/Ti interface and they share the same maximal translationengleiche subgroups *Amm2* and both of them are the minimal translationengleiche subgroups of *P63/mmc*), the evolution from hexagonal TiB to *Cmcm* TiB should be more favored in structure and symmetry. This is what we have obtained experimentally. We show more information below to address similar and further concerns of the referee.

Further comments

(4) P1, abstract, line 5 expression “close”: *The similarity to MAX phases (I agree) results not from a symmetry relation (although it can be discussed if one refers to the similarity and differences between octahedral surroundings and trigonal prisms). So I recommend another expression.*

Response: We thank the reviewer for pointing out this problem. We agree that the term of “close” is not precise enough. The sentence in the abstract is revised as “...**its space group was confirmed as $P\bar{6}m2$ (No. 187), which is in fact a hexagonal subgroup of $P6_3/mmc$ (No. 194), ...**”.

(5) P2, Discussion on TiB/Ti3B4: *see comments to Figure S1 (s. below).*

Response: We will respond below.

(6) Line 6: “3” *must be a subscript, please correct it within the whole manuscript and supporting information.*

Response: We thank the reviewer for pointing out this typo. We have corrected “*P63/mmc*” as “*P6₃/mmc*” through the whole manuscript.

(7) P2, second paragraph, discussion on MAX-/MAB-phases: *Just to focus on the orthorhombic metrics of the MAB phases is not sufficient for the distinction to the MAX phases. In MAB phases the topology gives 3 different characters for the connection of the trigonal prisms (in agreement to the orthorhombic structure) following the B-B-bonds, connection of the trigonal planes and perpendicular to the Al-layers. In the case of the MAX phases, the equivalence of a and b by symmetry is indeed a consequence of the symmetry.*

Response: We thank the reviewer for the constructive comments. Following the referee’s suggestion, in the revised manuscript, we have added more information to describe the difference of MAX- and MAB-phases. The corresponding part was revised to be: “**However, each boride presented in their work was orthorhombic, which is totally different from the hexagonal structures ($P6_3/mmc$ symmetry) of known MAX phases. In MAX phases, the M, A and X atoms alternately stack along a hexagonal close-packed (HCP) manner and respectively form equilateral triangles of their own (the equilateral nature is determined by the symmetry of the hexagonal space group) parallel to each other. However, M and A atoms in the reported borides alternately stack along orthorhombic manner. The M atoms, which are coordinated with boron, form non-**

equilateral trigonal prisms, with the side edge along x direction determining the lattice constant a and the non-equilateral M triangles perpendicular to the A layers. Furthermore, the nearest neighbor boron atoms form one-dimensional zig-zag chains perpendicular to the A layers also.” (from line 48 of page 2). We also revised the sentence of describing the structures of Ti_2InB_2 and Ti_2SnB_2 as “Because the B/Ti ratio (1.0) in Ti_2InB_2 and Ti_2SnB_2 is higher than those of X/M ratios (1/2, 2/3 or 3/4) in conventional MAX phases, boron atoms occupy the X sites between M layers and form a graphene-like layer (Supplementary Figs. 2 and 3) instead of a plane of equilateral triangles.”. (from line 121 of page 4)

(8) P2, third paragraph, classification of TiB and Ti_3B_4 as layer structures: All investigations on these type of compounds (for example *Acta Cryst. B*, 2015, 71, 777) show that there is a significant interaction between the transition metal atoms (also mentioned by the authors) and additionally via the seventh TM-B bond (for example in CrB: $4 \times 2.19 \text{ \AA}$, $2 \times 2.22 \text{ \AA}$, $1 \times 2.29 \text{ \AA}$) This is in contrast to the MAX-phases, where the interaction is much more anisotropic.

Response: We thank the reviewer for raising this discussion. First, we check the B-Ti distance in layered TiB structure and found the bond lengths are: $4 \times 2.35 \text{ \AA}$, $2 \times 2.37 \text{ \AA}$, $1 \times 2.52 \text{ \AA}$. The B-Ti bond length across the interlayer is much longer than those intralayer B-Ti bond lengths. Hence, it can be expected that the interlayer interaction in TiB can be weaker than that in CrB. Furthermore, we calculated the separation energies at different interfaces and found that the interaction at Ti-Ti interface (3.87 J/m^2) is similar with that at Ti-In interfaces in Ti_2InB_2 (3.27 J/m^2). And the strongest Ti-B separation energy in layered TiB is 8.26 J/m^2 and is comparable with that in Ti_2InB_2 (8.36 J/m^2). This indicates that the interaction in layered TiB is anisotropic also. We added the result of calculated separation energies of layered TiB in the Fig. S7 in the Supporting Information and added a sentence in the Sec. *Possibility of indium removal from Ti_2InB_2* : “The separation energy for the Ti/Ti interface of orthorhombic TiB was calculated to be 3.87 J/m^2 , which is comparable with that of Ti/In interface and indicates the laminated nature of TiB ($Cmcm$).” (from line 183 of page 7)

(9) P5, line 3: The reference to “stable” TiB and Ti_3B_4 ” is not clear. Which modification of is meant? $Pnma$ (existent) $Cmcm$ (not yet found in the binary system. Why is no reference to existing and stable TiB_2 , for example $2TiB + TiB_2 = Ti_3B_4$ (or Ti_2B_3 with the structure of V_2B_3 , or Ti_5B_6 ?)

Response: We thank the reviewer for pointing out this issue. In the binary search (the preliminary search), we can search all possible structures of A_xB_y from combination of ending compounds A and B within the limit of total atom number/cell. This means we can only use the energies of A and B as reference. In this search, we just specify the compositions of ending compounds instead of limiting the possible structures. So, both structures of $Pnma$ and $Cmcm$ phases of TiB were considered. In the next step, the stability of obtained Ti_2AB_2 or Ti_3AB_4 was confirmed in the global search by considering all possible compounds in Ti-B-A (A is Al, Sn, In, ...) ternary system including TiB_2 . We hope this explanation would be useful for the reviewer.

(10)P5, second paragraph: *Comment of a boron chemist: The structure chemistry of transition metal borides is dominated by trigonal prisms. Octahedral surroundings as they occur in MAX phases are extremely(!) rare. So a B-containing MAX—phase would be very surprising (comment: hexagonal borides like Ti_5Si_3B are not comparable, because the boron content is unknown, $Ti_5Si_3B_x$ ($x \sim 0.5??$)).*

Response: We thank the reviewer for this exciting comment. We agree that the B-containing MAX-phase is very rare and the conventional MAX-phases were limited to carbides and nitrides so far.

(11)P5, last paragraph, discussion on ELF and Bader charge. *I agree to the results on charge separation in Ti_2InB_2 . But the similarity to MAB phases and also binary transition metal borides requires, that results on these compounds are used for comparison (and cited) and not the results on boron-rich compounds.*

Response: We have added the ELF and Bader charge calculation results for TiB_2 and MAB phase Fe_2AlB_2 in the Supplementary Fig. 5(c). We revised the discussion in the manuscript as “**It is noteworthy to mention that the charge separation of Ti and B and B-B 2c-2e bonds in Ti_2InB_2 is close to the situation in TiB_2 (Supplementary Fig. 5). The boron atoms in MAB phase Fe_2AlB_2 arrange along B-B zig-zag chains through the formation of 2c-2e bonding.**”. (from line 155 of page 6)

(12)P6, Discussion on the stability of hexagonal TiB and orthorhombic TiB ($Cmcm$). *I definitely disagree to describe a transition of a “delaminated or dealloyed” Ti_2InB_2 , i. e. hypothetical hexagonal TiB , to TiB ($Cmcm$, i.e. CrB-type) as a lattice relaxation. There is a completely different structure with different bonding characteristics, that can’t be achieved by a lattice relaxation (I understand this term, that you have a change of the lattice parameter by some extent (15?) and therefore maybe a change of the crystal system).*

Response: We agree that the term “relaxation” is not precise in the description of phase change from hexagonal TiB to orthorhombic phase. We appreciate the referee for the constructive comments. Indeed, as the referee pointed out, lattice relaxation cannot drive the hexagonal TiB to the CrB-type TiB . The FPMD simulations show that the phase transition can happen under high temperature condition. Therefore, we have changed “lattice relaxation” to “strong thermal lattice vibrations” and revised the express in revised manuscript as “**Moreover, the first-principles molecular dynamics (FPMD) simulations revealed that a phase transition from hexagonal TiB to orthorhombic TiB compounds may occur due to the strong thermal lattice vibrations under high temperature conditions (Supplementary Fig. 10).**”. (from line 180 of page 7)

(13)P7, line 6, expression “6-member rings”: *The structure of Ti_2InB_2 contains no such rings of In and Ti atoms. The motif of Ti atoms and In atoms as well are from a closest packing (topologically a 36-net). The boron atoms form 6-member rings (63-net). I agree, that the projection of both atoms form hexagons.*

Response: We thank the reviewer for pointing out this problem. The “6-member ring”

was used by mistake to describe the projection of Ti and In atoms on the (001) plane (hexagons). Following the referee's comment, we have revised the related sentence as: "Observation along [001] direction provides an atomic image of Ti₂InB₂ in the x-y plane, with hexagonal patterns recording the projection of In and Ti atoms on the plane, which is consistent..."(from line 205 of page 7)

(14)P8, figure 3d: who is who? (but it doesn't matter, it's the same motif).

Response: We thank the reviewer for pointing out this problem. We added a description in the caption of Fig. 3d as "(blue and purple spheres respectively indicate Ti and In atoms)".

(15)P8/P9: discussion on the synthesis of quasi-2D MXene-like TiB: For this paragraph I have two questions. First, I do not understand the role of SiC. SiC is not known for its high affinity vs. oxygen. The surface of "several SiC ingots" seems not to be sufficient (in my opinion, but I haven't seen the samples). I'd suppose, that Mo has a higher affinity to O₂, especially, as MoO₃ is quite volatile. Furthermore, I don't know what the source of oxygen can be. So is there any proof, that SiC acts as a oxygen scavenger? For example, Zr sponge is really an oxygen scavenger (if this is an issue for the formation of TiB from Ti₂InB₂ by "dealloying"). Concerning the "dealloying" of In. Is there any experimental proof for the reaction? The second point is much more important for the interpretation of the results. I interpret the results as a decomposition of Ti₂InB₂ into metastable(?) TiB (CrB-type) release of In (evaporation? reaction?...). I don't agree, that a 2D-MXene-like product is formed. The P-XRD-pattern "as obtained" has more or less the same number of reflections with comparable line width as the other XRD patterns. For a real 2D structure like the defoliated MXenes I expect a XRD pattern with a strong emphasis of the reflections of the 2D layer (for example 00l for hexagonal TiB) and a significant broadening. This is not the case. So I see the formation of CrB-type TiB, that is quite well crystallized, although the particle size is smaller than the originally ones (as expected according to the formation from a decomposition).

Response: We thank the reviewer for raising this interesting discussion. Generally, SiC is used as an efficient oxygen scavenger under a relative high operation temperature because SiC is easy to react with O₂ as following equation: $\text{SiC} + \text{O}_2 \rightarrow \text{SiO}_2 + \text{CO}_2$. In our previous work (*Science*, **2011**, 333, 71), oxygen partial pressure (P_{O_2}) of $\sim 10^{-24}$ atm can be maintained when SiC was used by heating. Here, the oxygen comes from the quite limited diffused oxygen molecule from the air and also the adsorbed oxygen species in the wall of quartz tube. As shown in Figure S19, we also tried to use Ti to eliminate the oxygen traces during heating. Unfortunately, Ti powders were totally oxidized to TiO₂ during the reaction. Meanwhile, some TiO_x impurities were also generated in Ti₂InB₂ sample. We did not try other oxygen scavenger because SiC is efficient enough here.

For the dealloying process, we added the EDS and XRD results of the samples in the inner wall of the heated tube as indicated by the blue rectangle of Figure R1.

Figure R1. Schematic of TiB preparation through In evacuation. The blue rectangle indicates the deposited indium.

Figure R2. EDS (left) and XRD (right) spectra for the selected-area in Fig. R1.

Both characterizations in Fig. R2 indicates that the formed metal species in the inner wall of the quartz tube during the evacuation process was In metal, which directly confirmed the dealloying process of Ti_2InB_2 . And no TiB can be detected from the evacuated product in our device. As In was successfully removed from the parent Ti_2InB_2 , the characterizations of XRD and HRTEM on the obtained TiB fitted very well to our predicted TiB with orthorhombic space group ($Cmcm$). We agree with the referee that a real 2D MXenes structure was not obtained in the present research though the experimental obtained TiB ($Cmcm$) shows a layered (quasi-2D) character.

Moreover, our calculation confirms that the free energy of $Pnma$ TiB is always a little bit lower than the $Cmcm$ phase. Therefore, it is highly expected that $Pnma$ TiB could be the dominant product if a quick decomposition of Ti_2InB_2 happened. Thermodynamics will motivate the most stable phase (instead of a metastable CrB-type TiB) formed during a fast decomposition. Through a combing analysis of experiment and calculation results, we prefer to believe that the removal of In atoms from Ti_2InB_2 is not a result of a quite fast decomposition.

(16)P9, figure 4a and b: Fig 4a shows the formation of TiB between 975 and 1015°C. This increase is not related to a significant increase of the specific surface (4b). Therefore, I'd interpret these observations that the decomposition of Ti_2InB_2 occurs with a release of In and a quite fast formation of TiB ($Cmcm$) without intermediate hexagonal TiB. TiB ($Cmcm$) behaves not like a 2D material and is quite well crystallized (line width comparable to TiB_2). Therefore the "dealloyed" (in my view decomposed) Ti_2InB_2 can't be compared to a MXene. Please keep in mind that crystals of the CrB-type or ternary borides containing the structural unit of CrB

like MoAlB or Cr₂AlB₂ do not form very thin platelets like MAX-phases do. Furthermore, if one compares the SEM pictures of MXenes and figure 4c there is a significant difference. Obviously the transformation of the quite instable hexagonal TiB (no experimental proof for the occurrence as a metastable phase in the course of the decomposition despite the calculations) to TiB (Cmcm) results in microcrystalline samples of metastable (!) TiB.

Response: We accept the referee's criticism concerning the definition of the MXene on the obtained TiB. Indeed, we have no experimental proof of the intermediate hexagonal TiB. However, our FPMD simulation results gave the possibility of the space group of TiB changed from $P6\bar{m}2$ to $Cmcm$ mainly through an interlayer slip process. Here, the transformation of Ti₂InB₂ through hexagonal TiB to orthorhombic TiB was a hypothesis to describe the formation process of $Cmcm$ TiB because the hexagonal TiB was not stable compared with orthorhombic TiB under such a harsh reaction temperature. However, the $Cmcm$ TiB can be considered as a layered material because laminated units can be confirmed through SEM characterizations and the significantly different cleavage energies between Ti/Ti and Ti/B interfaces by calculations.

To our understanding, "Dealloying" refers to the selective removal of one element to form metastable compounds/structures from an alloy by heat treatment or corrosion processes. Actually, the production process of conventional MXenes by selective etching is also one kind of "dealloying". Normally, the "decomposition" is the separation of a single chemical compound into its two or more elemental parts or to simpler compounds. From the point view thermodynamics, the decomposition of Ti₂InB₂ can happen more or less at any temperature to reach a chemical reaction equilibrium. However, our synthesis process shows that Ti₂InB₂ can be very stable up to the temperature of 1200°C (Fig. S12). Therefore, we cannot expect that it would be decomposed completely in the temperature range of 900°C to 1050°C like shown in Fig. 4. Moreover, the layered structure of TiB was remained after the removal of In species during the high temperature treatment. Therefore, we feel that the production of $Cmcm$ TiB is mainly from the dealloying process but accompanied by only partial chemical decomposition, which induced the smaller particle size compared with its parent phase. Therefore, the term of "dealloying" could be more suitable to describe the whole process of TiB synthesis than "decomposition".

(17)P9, figure 4c: A comparison to Figure S19 shows, that significant amounts of carbon and oxygen represent the missing 20%. Therefore, the interpretation (or suggestion) of the analytical results as "single phase TiB" is too optimistic and requires a further explanation.

Response: We thank the reviewer for carefully reading the manuscript and providing this comment. We added following explanation in the notation of Supplementary Fig. 21 (Fig. 19 in the previous version): "The carbon signals in EDS results mainly originated from the carbon tape that we used to attach the powder sample onto the sample holder (copper stage). And the oxygen species should be the adsorbed oxygen molecule from the air because the TiB sample was exposed and stored in the air."

(18) I agree that Ti_2InB_2 represents a remarkable extension of the MAX phases by combination of the typical features of a transition metal boride (hexagonal TiB_2) with an intermetallic part ($TiIn$) following the principle of the MAX-phases and not the MAB-phases. The 2D character of Ti_2InB_2 is comparable to the MAX phases. I don't follow the authors in their conclusions the "dealloying" by thermal treatment at 975-1050°C results in a 2D structure of TiB ($Cmcm$) comparable to the MXenes. Therefore, the analogy concerning the application for example use as an anode material is not obvious as long as there is no experimental proof.

Response: We thank the reviewer for identifying Ti_2InB_2 as a new MAX phase instead of a MAB phase. We totally agree with the reviewer that $Cmcm$ TiB is not a MXene from the viewpoint of their 2D characters. We have not obtained hexagonal 2D TiB , a boron containing version of MXene, in the present work. We tried conventional soft route to remove In by using HF or LiF/HCl, but indium cannot be selectively etched. This is also a common problem for conventional In-containing MAX phases, and selective removal of In has not been reported so far. A suitable solvent is absolutely needed, and this is what we are trying to do in the next step. What we have done in the present study is the removal of In using a harsh route at high temperatures up to 1050 °C in vacuum. Surprisingly, In was completely removed and layered TiB ($Cmcm$, quasi-2D) arose as the main product. Although the obtained TiB ($Cmcm$) is not a MXene, it shares structure and symmetry similarities with the hypothetical TiB MXene as we mentioned earlier (both of them have 2D characters with a space separating the Ti/Ti interface and they share the same maximal translationengleiche subgroups Amm_2). Therefore, we suggest that the obtained TiB ($Cmcm$) is a high-temperature phase of the hypothetical hexagonal TiB , and it is strongly indicative of the possibility to obtain TiB MXene at mild conditions. As a highly possible family member of MXenes, the properties of TiB monolayer are fascinating us and can be very interesting to scientific communities. Therefore, it should be of both scientific and technological importance to have a detailed understanding of its basic properties and potential applications of 2D TiB .

Following the reviewer's comments, we make the situation that the obtained TiB ($Cmcm$) is not comparable to MXenes clear in the revised manuscript. We revised the description through the manuscript to change "MXene-like (quasi-2D) TiB " to "layered TiB ". Furthermore, we inserted one sentence in Sec. **Prospect of 2D TiB as anode material for Li/Na ion batteries** "Although the layered TiB ($Cmcm$) is not comparable with 2D MXenes, the successful removal of In atoms suggests the possibility to obtain TiB MXene at mild conditions." (from line 289 of page 11)

Supplementary information:

(19) Please change the space group symbol of " $P-6m2$ " the letter " M " to a " m ".

Response: We thank the reviewer for pointing out this problem. We have corrected the symbol through the whole manuscript.

(20) Figure S1: In my opinion it is misleading to discuss only TiB ($Cmcm$) as a result of

the calculations (self-fulfilling prophecy). There is no comment on the stability of experimentally established TiB (Pnma) (Acta Cryst 1954, 7, 77; Monatsh. Chem. 1960, 91, 608;) Furthermore, I miss a reference to the experimentally established compound Ti₃B₄. Both aspects should be considered for the discussion on P2 and the introduction of the manuscript. Comment from an experimentalist: The existence of a compound is not proven by calculation of a local minimum in the energy hypersurface and vice versa its non-existence by the absence of this minimum.

Response: We thank the reviewer for this valuable suggestion. We have calculated the free energy of the two phases (*Pnma* and *Cmcm*) of TiB. The calculations give a consistent result with experiment that *Pnma* phase is more stable than *Cmcm* phase. The phonon dispersion calculation shows no imaginary frequency for *Cmcm* phase, which indicate *Cmcm* TiB can be metastable phase and is synthesizable under certain condition. We have added two references (Refs. 23 and 24) about the experimental synthesis of TiB and Ti₃B₄ and modified related expression in the introduction part in the revised manuscript: “**The predicted Ti₃B₄ shows the same structure as that reported in the previous experiment²³. However, the reported TiB compound in experiment²⁴ possesses the space group of *Pnma*, indicating that a direct formation of layered TiB (*Cmcm*) is prohibited.**” (from line 67 of page 2)

(21)Table S1: Position/coordinates/site of B in the models of space group P-6m2: In my version of the International Tables for Crystallography the site 0,0,0.5 (Wyckoff 1b) is a onefold site. Therefore, the compositions given as TiB or Ti₂InB₂ should be Ti₂B and Ti₂InB, respectively. If the authors want to describe the structure model as discussed in the manuscript they have to add a second boron site (I suggest Wyckoff site 1f 2/3,1/3,1/2). For the case, that there is really only one site for boron the complete discussion including the calculations of the stability must be completely changed (note: XRD-patterns will not be affected).

Response: We thank the reviewer for this professional comment. The structure we used for structural analysis and theoretical calculations indeed contains two boron sites: one is Wyckoff 1b site (0, 0, 0.5) and the other is Wyckoff 1f site (2/3, 1/3, 1/2), as the reviewer pointed out. In the revised supporting information, we added another boron site (Wyckoff 1f) to the Table S1.

REVIEWERS' COMMENTS:

Reviewer #1 (Remarks to the Author):

The authors have adequately taken my comments into account. The paper can be published now.

Reviewer #2 (Remarks to the Author):

I have gone through the revised version and happy to see that authors have incorporated the changes suggested. I recommend the acceptance of the manuscript in its present form.

Reviewer #4 (Remarks to the Author):

In general, I am very impressed of the research level of this work, especially of a perfect balance between its theoretical and experimental parts. The authors have convincingly demonstrated how modern methods of structure prediction can help in discovering new crystalline substances with non-trivial structure and composition. It is not surprising that such rich material raised a lively discussion and provoked many questions of the reviewers. As a crystallographer, I was particularly interested in the crystallochemical part of the discussion. I can conclude that all issues concerning symmetry and topology of the phases were rightly raised by the reviewers, but the authors have correctly addressed all of the issues. I think the revised manuscript can be published in Nature Communications in its current form.

Point-by-point response to reviewers' comments:

We would like to thank the three reviewers for their very positive recommendation.

Reviewer #1 (Remarks to the Author):

The authors have adequately taken my comments into account. The paper can be published now.

Reviewer #2 (Remarks to the Author):

I have gone through the revised version and happy to see that authors have incorporated the changes suggested. I recommend the acceptance of the manuscript in its present form.

Reviewer #4 (Remarks to the Author):

In general, I am very impressed of the research level of this work, especially of a perfect balance between its theoretical and experimental parts. The authors have convincingly demonstrated how modern methods of structure prediction can help in discovering new crystalline substances with non-trivial structure and composition. It is not surprising that such rich material raised a lively discussion and provoked many questions of the reviewers. As a crystallographer, I was particularly interested in the crystallochemical part of the discussion. I can conclude that all issues concerning symmetry and topology of the phases were rightly raised by the reviewers, but the authors have correctly addressed all of the issues. I think the revised manuscript can be published in Nature Communications in its current form.